# Optimal Resource Allocation for Early Stopping-based Neural Architecture Search Methods

Marcel Aach[1,2]  Eray Inanc[1]  Rakesh Sarma[1]  Morris Riedel[1,2]  Andreas Lintermann[1]

[1]Jülich Supercomputing Centre, Forschungszentrum Jülich, Germany
[2]School of Engineering and Natural Sciences, University of Iceland, Iceland

**Abstract**  The field of Neural Architecture Search (NAS) has been significantly benefiting from the increased availability of parallel compute resources, as optimization algorithms typically require sampling and evaluating hundreds of model configurations. Consequently, to make use of these resources, the most commonly used early stopping-based NAS methods are suitable for running multiple trials in parallel. At the same time, also the training time of single model configurations can be reduced, e.g., by employing data-parallel training using multiple Graphics Processing Units (GPUs). This paper investigates the optimal allocation of a fixed amount of parallel workers for conducting NAS. In practice, users have to decide if the computational resources are primarily used to assign more workers to the training of individual trials or to increase the number of trials executed in parallel. The first option accelerates the speed of the individual trials (exploitation) but reduces the parallelism of the NAS loop, whereas for the second option, the runtime of the trials is longer but a larger number of simultaneously processed trials in the NAS loop is achieved (exploration). Our study encompasses both large- and small-scale scenarios, including tuning models in parallel on a single GPU, with data-parallel training on up to 16 GPUs, and measuring the scalability of NAS on up to 64 GPUs. Our empirical results using the HyperBand, Asynchronous Successive Halving, and Bayesian Optimization HyperBand methods offer valuable insights for users seeking to run NAS on both small and large computational budgets. By selecting the appropriate number of parallel evaluations, the NAS process can be accelerated by factors of $\approx 2 - 5$ while preserving the test set accuracy compared to non-optimal resource allocations.

## 1 Introduction

Neural Architecture Search (NAS) describes the process of automatically designing neural networks for various applications (Hutter et al., 2019). As NAS requires searching through a vast space of possible neural network architectures to find the one that best suits the given problem, this process is computationally expensive. Therefore, there has been a growing interest in exploiting parallel computing to speed up this process. By using parallelism, multiple candidate architectures can be evaluated concurrently, which significantly reduces the total runtime to find an architecture candidate that performs well. In recent years, the available NAS and Hyperparameter Optimization (HPO) frameworks have made an effort to run in parallel (Li et al., 2020a; Kandasamy et al., 2020; Liaw et al., 2018; Balaprakash et al., 2018). However, investigations into how these methods scale up are sparse and the optimal degree of parallelism to use in the NAS process – when only limited resources are available – is still an open question.

This paper aims to investigate the impact of the parallelism of the NAS loop on the runtime and quality of the outcome of three of the most commonly used early stopping-based NAS methods: Hyperband (HB) by (Li et al., 2017), Asynchronous Successive Halving Algorithm) (ASHA) by (Li et al., 2020a), and Bayesian Optimization and HyperBand (BOHB) by (Falkner et al., 2018). All of these algorithms have 'meta-hyperparameters' themselves, that have a large influence on the

performance. One example is the reduction factor, which defines the percentage of trials that are terminated by early stopping. Investigations have shown that a reduction factor of three or four is a robust choice (Li et al., 2020a). However, the optimal number of parallel evaluations that should be used has not been studied extensively.

We study the effect of parallelism on the NAS methods using both, small and large computational budgets on the standardized NATS-Benchmark (Dong et al., 2021), training up to 256 Convolutional Neural Networks (CNNs) on three image classification datasets. In the first step, the optimal number of parallel evaluations to run when using a **single** Graphics Processing Unit (GPU) – also refereed to as a worker or a device in the following – is evaluated. As a second step we increase the number of GPUs to 16 and vary the number of parallel evaluations by running data-parallel training of the architectures on one to four GPUs per trial. Our main objective is to explore the balance between accelerating single trials by allocating more resources per trial and speeding up the entire NAS loop by running more trials in parallel, considering a fixed amount of available hardware resources. The original motivation of HB and its successors ASHA and BOHB was to develop an adaptive, multi-fidelity approach for resource allocation, however, mainly from a temporal perspective. That is, 'resources' or fidelity are defined as a fixed amount of training epochs a NAS run can use and that then is shared between trials.

In this work, 'resources' and fidelity are defined as fixed amount of physical workers to address this issue from a hardware perspective. To investigate the **efficient utilization** of these workers, we are looking to answer the following two research questions:

- **RQ-1**: How many resources should be allocated in total for a NAS?
- **RQ-2**: How many resources should be allocated for each NAS trial?

Our main contribution is an investigation of these research questions on small- and large-scale computational budgets (up to 64 GPUs), featuring different algorithms, types and sizes of datasets, and neural networks and types of GPUs. Our findings suggest to increase the parallelism of the NAS loop at the cost of the parallelism of the single trials, independent of the factors such as algorithm, dataset, or hardware type.

In Sec. 2, we introduce the methods used for evaluation and embed this study in related scalability and resource allocation work. Section 3 then describes the benchmark and technical frameworks used, while in Sec. 4 and Sec. 5 the results of the experiments on various scales are reported and discussed. We finish with a conclusion in Sec.6.

## 2 Related Work

### 2.1 Hyperband (HB)

In Machine Learning (ML), the performance of a certain model with respect to a certain metric, i.e., the validation error, can be described by the function $f : \mathcal{X} \to \mathbb{R}$ where $\mathcal{X}$ is the space of possible model architectures. The main goal of NAS is to minimize the objective function $f$ by finding an architecture configuration $x^* \in \mathcal{X}$ such that $x^* \in \arg\min_{x \in \mathcal{X}} f(x)$. Evaluations of the objective functions are expensive as the model configurations usually require full training. HB (Li et al., 2017) seeks to approximate this objective function by evaluating it on a smaller budget, e.g., by running a trial with fewer epochs. Worse performing trials are then cut off early and their resources are transferred to the best performing trials. This early stopping procedure is known as Successive Halving (Jamieson and Talwalkar, 2016). To assess the most promising trials, HB waits for *all* trials of a bracket to reach a certain threshold in time before applying Successive Halving. This leads to idling workers as some will be faster than others (which can be circumvented by spawning a new bracket if enough workers are available). HB is a pure scheduling algorithm as new configurations to evaluate are randomly sampled.

## 2.2 Bayesian Optimization Hyperband (BOHB)

Another option to accelerate the NAS process is the use of optimization methods to find the minimum of $f$, where Bayesian optimization (BO) has become the main method of choice. The idea behind BO is to use a probabilistic model of $f$ based on data points observed in the past. In the case of NAS, this means finding promising new model architectures based on the performance of past trials.

BO is computationally costly and BOHB (Falkner et al., 2018) solves this issue by combining the optimization process with HB for scheduling. In BOHB, HB chooses the number of hyperparameter configurations and their assigned budget. BO chooses the hyperparameters by deploying a tree parzen estimator (Bergstra et al., 2011). Combining both approaches has several advantages: good results are achievable on smaller budgets while on larger budgets the performance is better than other methods, such as plain HB or random search (Bergstra and Bengio, 2012).

## 2.3 Asynchronous Successive Halving Algorithm

ASHA (Li et al., 2020a) addresses the problem of large scale NAS by improving the scalability of HB. To avoid idling workers ASHA, in contrast to HB decides on a rolling basis which trials are promising. When two trials reach the time barrier, the trial with the better performance is continued while the other is paused until the performance of the next completed trial can be juxtaposed. This asynchronous comparison leads to speed-ups. ASHA focuses purely on the scheduling aspects of NAS and new architecture candidates are randomly sampled, contrary to BOHB.

## 2.4 Scalability and Resource Allocation Investigations

In the original works on ASHA and BOHB, initial investigations on their scalability with respect to the total number of workers used in the NAS process are already performed. In the BOHB case, the algorithm is scaled up to 32 Central Processing Unit (CPU) workers to perform HPO on a small benchmarking dataset from OpenML (Frey and Slate, 1991; Vanschoren et al., 2014), attaining a speed-up of $15x$ in comparison to using a single worker. However, no scalability results on larger datasets or GPUs are reported. In a proposed extension to the BOHB framework, substantial speed-ups are measured, still just on small multi-layer perceptron models on tabular datasets (Klein et al., 2020).

While in the ASHA paper, a study on a total number of 500 GPUs is reported, the authors do not juxtapose these results with results achieved on a smaller number of workers for the same problem to compute a speed-up. The paper already addresses the trade-off of using resources to train a model faster vs. evaluating more models in parallel. This is, however, done on just a single model and dataset, and by simulating the training times using an analytical performance model (Qi et al., 2017). In contrast, our study performs evaluations on different types of GPUs, encompasses larger models and datasets, and takes measurements by actually performing the training runs (instead of simulating them).

## 2.5 Data-Parallel Deep Learning

Data-parallel training is especially suited for training deep neural networks on large datasets on multiple GPUs. This method greatly accelerates the training process.

To perform data-parallel training, the training dataset is partitioned across multiple workers, and each worker trains a copy of the model on a subset of the data. During each iteration, the gradients of the loss function with respect to the model parameters are computed on each device, and then averaged across all devices to obtain a global gradient update. The model parameters are then updated using this global gradient (Li et al., 2020b). The global gradient update can be computed as

$$\Delta w = \frac{1}{N} \sum_{i=1}^{N} \Delta w_i, \tag{1}$$

where $\Delta w_i$ is the gradient computed on device $i$ and $N$ is the total number of devices. With a growing number of devices, this averaged gradient changes, which impacts the generalization performance (Keskar et al., 2017). It is possible to circumvent this degradation of performance by scaling the learning rate with the number of devices (Goyal et al., 2017). This works, however, only if the number of devices that is trained on in parallel is small. Another way to avoid this degradation is to optimize the learning rate using BO (Égelé et al., 2021).

## 3 Methodology

### 3.1 Ray Tune

For our experiments we use the open-source Ray Tune (Liaw et al., 2018) library. It offers to run multiple NAS optimizations across multiple GPUs via a unified interface. We use the implementations of ASHA, BOHB, and HB from within Ray Tune to eliminate implementation-specific disturbance factors in our analysis. All communication and scheduling is therefore handled via Ray. One of the most important features of the Ray library is the possibility to easily modify the number of workers to use per evaluation, even allowing for floating point values. This way, it is possible to run multiple trials in parallel on a single GPU, which share its compute power via 'context swichting'. In our experiments, we use Ray Tune for the NAS loop and PyTorch-DDP (Li et al., 2020b) for the training of the single models.

### 3.2 NATS Benchmark

Benchmarks play an important role in NAS research by providing a standardized evaluation framework for comparing different NAS algorithms on multiple datasets. For this study, we run our main experiments on the size search space $S_s$ from the NATS-Bench by (Dong et al., 2021). For this size search space, a general CNN model with fixed topology is created, where the number of channels for each stage of convolutional layers is sampled from $k \cdot 8, k \in \{1, \ldots, 8\}$. As there are five layer stages in the model, the total number of possible configurations is $8^5$. We train all models with the hyperparameters mentioned in the original NATS-Bench paper. This includes the usage of the Stochastic Gradient Descent optimizer with learning rate of $lr = 0.1$, momentum of $mom = 0.9$, weight decay of $wd = 0.0005$, batch size of $bs = 256$, and nesterov momentum for 90 epochs. The quantity $lr$ is annealed from 0.1 to 0 over the course of training with the cosine annealing schedule (Loshchilov and Hutter, 2017), also in accordance with the NATS-Bench training protocol.

### 3.3 Datasets

We train each model configuration on the most popular image classification datasets cifar-10, cifar-100 (Krizhevsky, 2009), and imagenet (Russakovsky et al., 2015). To reduce the computational costs of our analysis, we use a down-sampled variant of imagenet for most of our experiments, where each image is scaled to $16 \times 16$ pixels and the number of classes is reduced to 120 (also referred to as imagenet16-120). As these datasets do not feature a validation dataset, we set aside 20% of the training dataset for validation to evaluate the obtained model configurations during the NAS optimization process[1]. The testset remains untouched. For all datasets, we reuse the pre-processing and augmentation techniques from NATS-Bench.

## 4 Experimental Results

In this section, the experimental results are presented, evaluating the scalability of the different NAS methods on up to 64 GPU and the general performance of NAS methods when sharing a single or

---

[1]This is different from the dataset splits reported in NATS-Bench, which uses a 50/50 split. However, to measure the effect of resource allocation, we need our training dataset to be sufficiently large, which is why we chose the 80/20 split.

multiple GPUs per trial. The experiments from Sec. 4.1, 4.2, 4.3 and 4.4 are performed on JURECA-DC-GPU (Jülich Supercomputing Centre, 2021), a High-Performance Computing (HPC) machine that features NVIDIA A100[2] GPUs, while the experiments from Sec. 4.5 are performed on the MareNosturm4 CTE-AMD machine, equipped with AMD MI50[3] GPUs and the DEEP-EST (Suarez et al., 2021) machine, with NVIDIA V100[4] GPUs. Code to reproduce the results is available via a GitLab repository[5].

### 4.1 General Scalability

To answer the first research question (**RQ-1**), this section investigates the general scalability of the three early-stopping based NAS methods, plain HB, BOHB, and ASHA. As this work focuses on efficient resource allocation, the question at hand is to find out at which point adding more workers to the NAS task is not profitable anymore to guarantee efficient utilization. A common metric for tracking this is to perform strong scaling tests and compare the parallel efficiencies $E_G$ of different NAS methods. In strong scaling analyses, the problem size remains constant while the number of workers is increased. In this case, the problem size is defined as the total number of samples the NAS methods need to evaluate, and is set to 128. The speedup can be calculated by

$$S_G = \frac{T_1}{T_G}, \tag{2}$$

where $G$ is the number of GPUs, $T_1$ is the runtime of the single GPU baseline case, and $T_G$ is the runtime on $G$ GPUs. Then, $E_G$ can be expressed by

$$E_G = \frac{S_G}{G}, \quad E_G \in [0, 1] \tag{3}$$

In case $E_G$ equals (or is close to) unity, the best possible scaling performance is achieved (only in exceptional cases values larger than unity are possible). Analyzing $E_G$ allows for a standardized comparison of the different methods. Figure 1 shows $E_G$ values for setups between 1 and 64 GPUs on all three datasets from NATS-Bench, running one trial per GPU. The random search (RAND) plots serve as a best case scenario as all trials are fully trained independently from each other and there is no early stopping. Allocation of more GPU resources reduces the runtime almost proportionally. Hence, $E_G$ values for this case are always above 0.85. Among other methods, the ASHA case achieves the best parallel performance on up to 8 GPUs with $E_G = 1$ across all three datasets. This result is expected, since the trials in this method run in asynchronous manner. Overall, all methods perform reasonably well up to 32 GPUs with $E_G$ values above 0.75. However, it can be seen that $E_G$ drops below 0.75 on 64 GPUs across all datasets and algorithms. The drop in performance in comparison to random search becomes apparent for larger number of GPUs.

### 4.2 Resource Allocation on a Single GPU

To answer the second research question (**RQ-2**), we investigate how the amount of resources allocated to a single trial impacts the total runtime of the NAS process.

For the experiments depicted in Fig. 2, the different early stopping-based NAS algorithms run until 64 different architecture candidate samples have been evaluated. The plots depict the validation accuracy of the best architecture currently discovered by the NAS over the time for running up to eight trials in parallel on a single GPU. The plots are partly truncated on the x- and y-axis, and standard deviation is omitted for readability reasons, see App. A for full plots including

---

[2]https://www.nvidia.com/en-us/data-center/a100/

[3]https://www.amd.com/en/products/professional-graphics/instinct-mi50

[4]https://www.nvidia.com/en-us/data-center/v100/

[5]https://gitlab.jsc.fz-juelich.de/CoE-RAISE/FZJ/nas-allocation

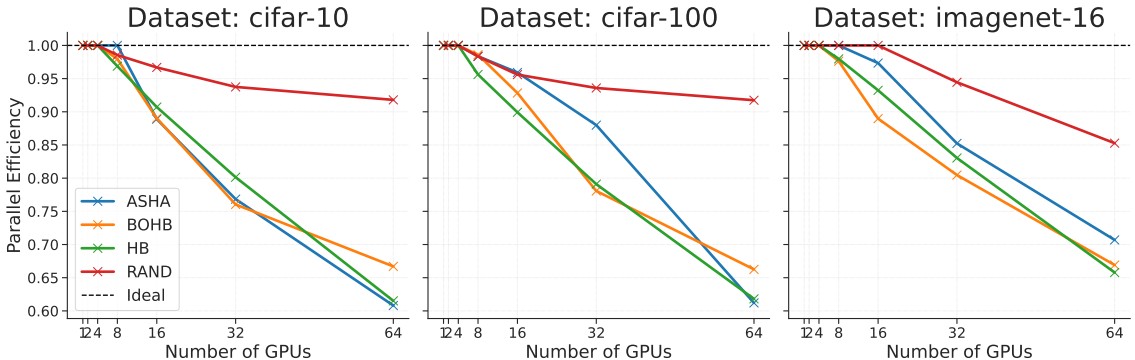

Figure 1: Scalability comparison: Parallel efficiency $E_G$ over the number of GPUs, based on the runtime until 128 samples are evaluated. $E_G = 1$ denotes the ideal case.

error bars. Comparing the ASHA, BOHB, and HB to the random search (RAND) runtimes in Fig. 2 proves that early stopping methods greatly benefit the computation. The differences between the final validation accuracies that the NAS methods converge to are with less than 1% only marginal across all datasets. BOHB is the only exception, but also here differences are only slightly above 1%. Therefore, it can be stated that the validation accuracy is not impacted by the number of parallel trials.

It is, however, evident that the runtime is largely influenced by the parallelism of the NAS algorithm, i.e., the NAS methods take up to twice as long to finish when running just a single trial at a time (blue lines in Fig. 2). The main reason for this is that early stopping-based methods can only make their decisions on which trials to stop and which to continue based on having multiple, intermediate results available. If just one trial is running (sequentially) at a time, the algorithm needs to constantly checkpoint the current trial, load a new architecture, and train it for a specified amount of time to have a comparison point. This checkpointing and re-loading creates overhead that ultimately leads to longer runtimes. When running multiple trials in parallel, these kinds of decisions can be made on the fly. While the amount of checkpoint operations is similar, less re-loading is necessary in this case. The differences between the runtimes for 2, 4, and 8 trials per GPU are small (less than 2,000s). For both, the ASHA and BOHB case running 4 or 8 trials in parallel take roughly the same time to finish. What is interesting to observe is the fact that the *fewer* resources are used per trial, i.e., 1, 2, or 4 trials per GPU, the *faster* the NAS method discovers architectures with high validation accuracies, at least at the beginning. This can be observed by the steep increase of the blue, orange, and green lines in Fig. 3 for small runtimes. This results in enhanced anytime performance for the NAS loop, i.e., stopping the algorithm at a random point in time and evaluating the performance at that point. This means that halting the process at a random moment would likely produce higher validation accuracy than when running the maximum of 8 trials per GPU.

In addition to the frequent checkpointing and re-loading, one possible reason for the longer runtime of the sequential trial case could be the idling of the GPU due to not receiving sufficient data for calculations fast enough. When running 8 trials in parallel, the GPU processes $8 \cdot 256 = 2{,}048$ data samples at a time, while in the single trial case just 256 samples (=one batch) are processed at a time. The latter is not enough to efficiently utilize a GPU.

To account for this effect, a reference experiment is conducted, where the quantity $bs$ is scaled inversely with the number of trials running on the GPU. Figure 3 shows the corresponding results. Usually, a larger $bs$ value leads to a faster training time, as the GPU can streamline the computations of the gradients better and does not have to interrupt as often to perform an optimizer update. For

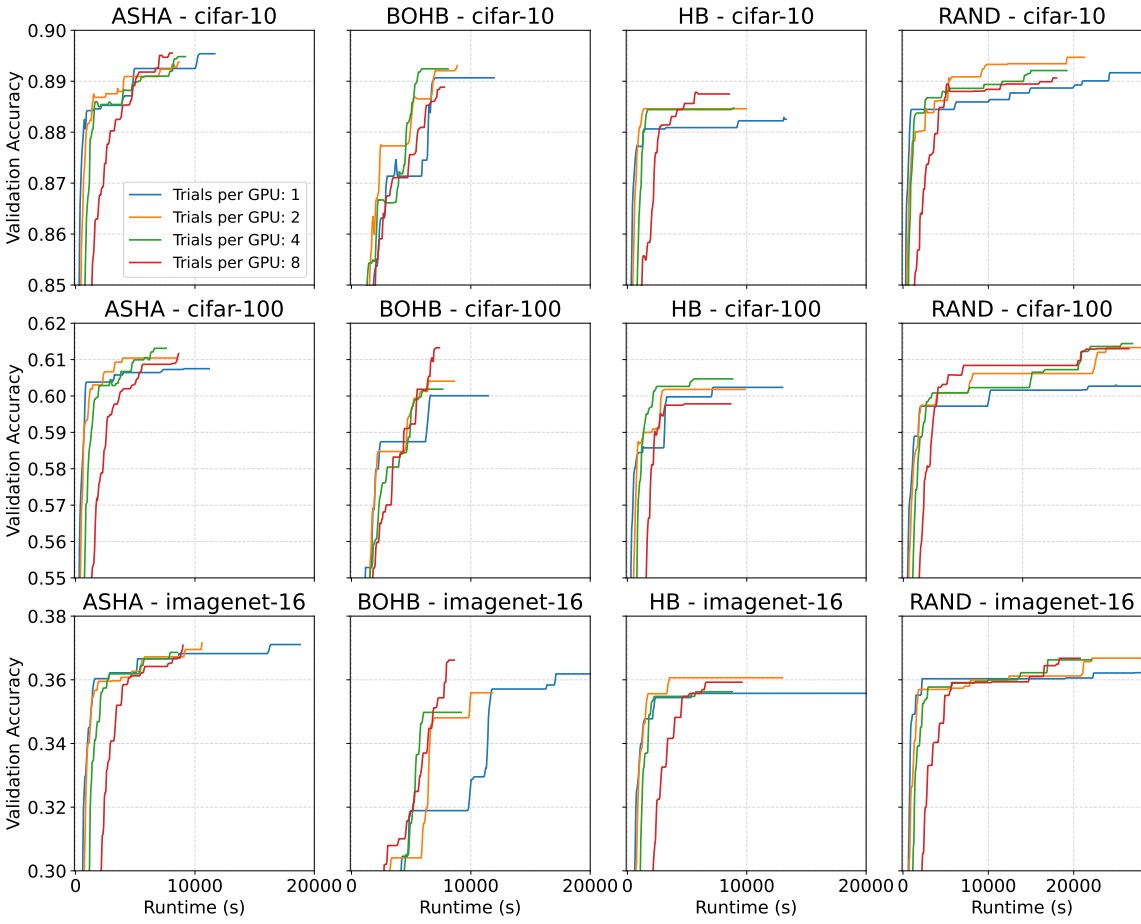

Figure 2: Multiple trials on a single GPU, showing validation accuracy over runtime. Results are averaged over three different seeds. The plots are partly truncated for readability. In all cases, running 1 trial per GPU (blue line) clearly has the longest runtime while running 4 or 8 trials (green and orange lines) per GPU finish faster.

this test, the batch size is set to $bs = 256$ as the baseline for the '8 Trials per GPU', then $bs = 2 \cdot 256$ is selected for the '4 Trials per GPU' case, and so on. By doing this, we align the number of data samples processed between the different levels of parallelism. To obtain comparable performance in terms of validation accuracy, the quantity $lr$ is scaled with the batch size. We only run this test for the imagenet-16-120 dataset as the linear batch size and $lr$ scaling already degrades the validation accuracy when training on the cifar-10 and cifar-100 datasets.

The results show that the time difference between the '1 Trial per GPU' scenario and the other cases reduces significantly. Table 1 compares the outcomes and GPU utilizations of the NAS process with and without the batch scaling for the ASHA case. While the factor in average runtime between non-optimal ('1 Trial per GPU') and optimized ('4 Trial per GPU') resource allocation with no batch scaling is at $18,817s/8,524s \approx 2.2$, it reduces to $10,749s/6,989s \approx 1.54$ for the batch scaling ('1 vs. 2 Trials per GPU'). The batch scaling also improves the GPU utilization of almost all resource allocation strategies, however, for the sequential scenario it only increases from 38% to 49%. This finding confirms that the main reason hindering efficient resource usage is the switching and reloading processes between trials and not the amount of simultaneously processed data. Overall it can again be seen that the resulting test set accuracy is not impacted by the number of parallel trials.

Table 1: Averaged results of using ASHA on the imagenet-16 dataset on a **single** GPU with standard deviation as uncertainty measurement. The runtime is the wallclock time of the whole NAS loop. The test set accuracy is computed by using the architecture with the best performance on the validation set. GPU utilization is measured using an internal tool of the HPC machine.

| Trials per GPU | Without (inverse) batch scaling | | | With (inverse) batch scaling | | |
|---|---|---|---|---|---|---|
| | Runtime | Test set accuracy | GPU utilization | Runtime | Test set accuracy | GPU utilization |
| 1 | 18,817 ±414 s | 37.6 ±0.77% | 38 ±17 % | 10,749 ±257 s | 36.7 ±0.24 % | 49 ±21 % |
| 2 | 10,583 ±212 s | 37.7 ±1.09% | 79 ±9 % | 6,989 ±248 s | 37.3 ±0.11 % | 82 ±5 % |
| 4 | 8,524 ±113 s | 37.7 ±0.91% | 83 ±5 % | 7,495 ±230 s | 37.1 ±0.49 % | 82 ±4 % |
| 8 | 8,996 ±152 s | 37.5 ±0.30% | 86 ±3 % | 9,252 ±237 s | 37.4 ±0.06 % | 86 ±4 % |

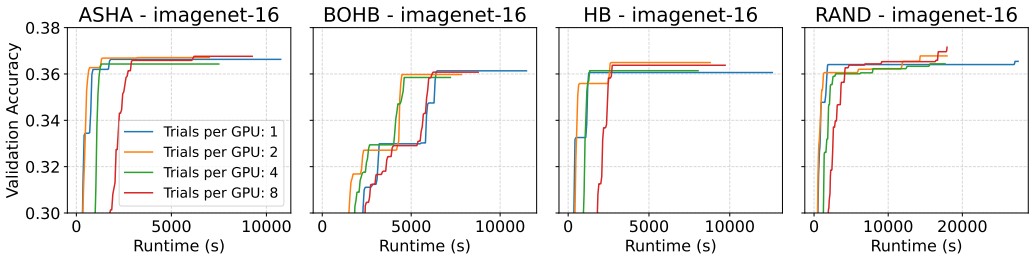

Figure 3: Multiple trials on a single GPU with inverse batch scaling according to the number of trials, showing validation accuracy over runtime.

## 4.3 Resource Allocation on Multiple GPUs with Data-Parallel Training

The effect of employing a higher number of workers per trial is investigated here to answer the second research question (**RQ-2**). Therefore, data-parallel training, see Sec. 2.5, is employed. To obtain the results depicted in Fig. 4, the runtime of the early stopping-based NAS algorithms on a **fixed** number of 16 GPUs is measured to evaluate 64 different architecture candidates. The base case with a batch size of $bs = 256$ and a learning rate of $lr = 0.1$ is scaled linearly with the number of GPUs.

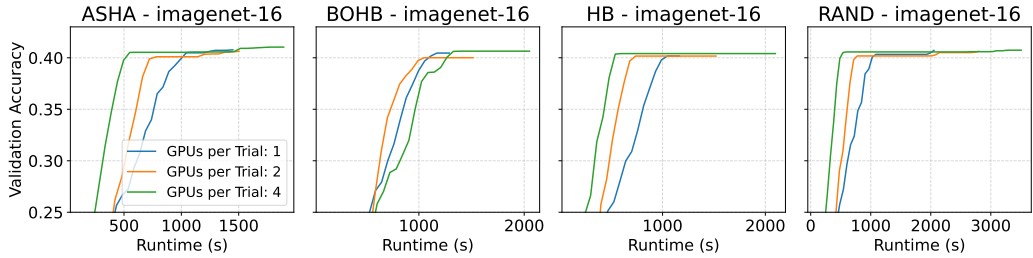

Figure 4: Distributing trials across **multiple** GPUs, showing validation accuracy over runtime. The results are averaged over three different seeds.

The results are in line with the findings from the single GPU case. When using four GPUs per trial, the NAS methods take the longest time to finish. For the ASHA and BOHB cases, this implies that the runtimes of the 'GPUs per trial: 1' configuration are ≈ 1.3 and ≈ 1.59 faster than the 'GPUs per trial: 4' configuration. However, with the exception of BOHB, the observations regarding the anytime performance hold true here. That is, using more GPUs per trial accelerates the discovery of architectures with better validation accuracies in the beginning of the NAS run.

To investigate resource allocations on a larger computational budget, a NAS study for the ASHA algorithm is performed with a fixed number of 64 GPUs as resources, measuring the runtime until 256 model architectures are evaluated. The number of workers to use for the data-parallel training is varied from $1 - 16$. The results are depicted in Fig. 5. Again, it is observed that increasing the parallelism of the NAS loop and decreasing the parallelism of the data-parallel training ('GPUs per trial: 1 or 2' case) results in an optimal resource allocation. In this case, the factor between optimized and sub-optimal allocations becomes even larger at $13{,}711s/2{,}601s \approx 5.2$ with no impact on the test set accuracy.

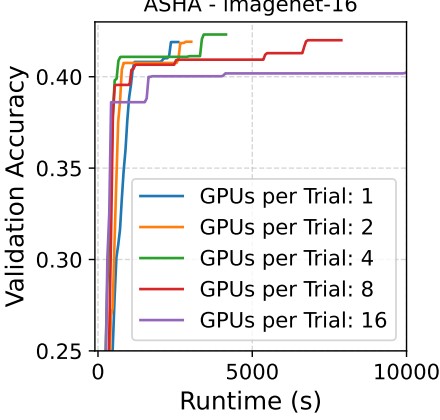

| GPUs per trial | Runtime | Test set accuracy |
|---|---|---|
| 1 | 2,601 s | 43.4 % |
| 2 | 3,021 s | 42.7 % |
| 4 | 4,148 s | 42.3 % |
| 8 | 7,886 s | 42.9 % |
| 16 | 13,711 s | 41.4 % |

Figure 5: Distributing trials across a total of 64 GPUs for ASHA on imagenet-16. Left (plot): The validation accuracy over runtime until 256 samples are evaluated. The plot is partly truncated for readability. Right (table): Overview of runtime and test set accuracy achieved.

### 4.4 Evaluation with Large Models

To test how well the findings transfer to larger neural network models and datasets, we perform an additional evaluation with a custom ResNet50 (He et al., 2016) search space, where the number of convolutional filters at each stage of the ResNet is sampled from the set $\mathcal{S} = \{64, 128, 256, 512\}$. As there are four layer stages in the model, the total number of possible configurations is $4^4 = 256$. The sampled models are roughly two orders of magnitude larger in terms of the number of parameters than the models from the NATS-Bench search space (e.g., 30M vs. 0.3M parameters). The training is performed on the complete imagenet-1k dataset ($\approx 300$ GB in size), utilizing the same training protocol as for the NATS-Bench experiments and using a fixed budget of 64 GPUs. The results in Fig. 6 (left) show that also in this case increasing the parallelism of the NAS loop at the cost of the parallelism of the single trials reduces the total runtime significantly (blue vs. green line).

### 4.5 Evaluation on Different GPU Types

To asses to what extend the results are influenced by the type of GPU utilized for the NAS runs, we perform additional experiments on different HPC machines, featuring less powerful GPUs than the NVIDIA A100s from the previous sections. For the single GPU case (Sec. 4.2), one AMD MI50 is used while for the multi-GPU case (Sec. 4.3) 16 NVIDIA V100s are used for a comparison. The results are depicted in Fig. 6 (middle and right). For both experiments, it can be observed that running more trials in parallel leads to a reduced runtime (red line in the middle plot and blue line in the right plot).

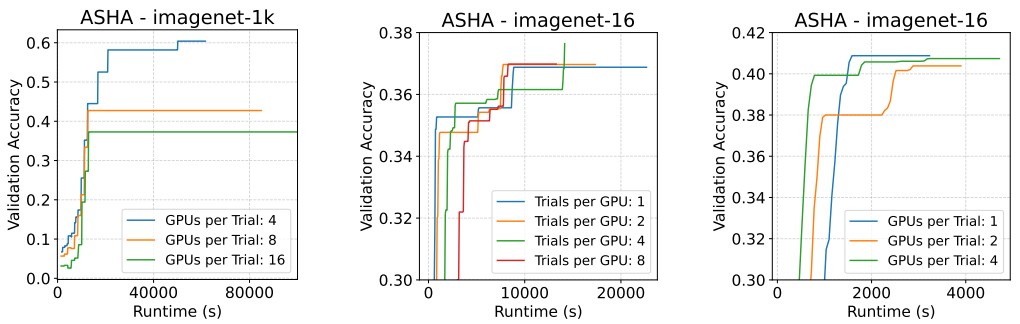

Figure 6: Left: Running trials with ResNet search space on imagenet-1k. Middle: Running trials on a single AMD MI50 GPU. Right: Running trials across 16 NVIDIA V100 GPUs.

## 5  Discussion and Insights

From the results, several things become evident. First, to answer **RQ1**, the general scalabilty of early stopping-based NAS methods show good performance on up to 32 GPU workers in parallel. This is not only true for the smaller cifar-10 and cifar-100 datasets, but also for the larger imagenet 16-120 dataset. A good scalability indicates that adding more workers to a problem proportionally reduces the total runtime, leading to faster computation of the NAS loop and optimized resource usage. The benefits of adding more GPUs to a problem outweighs the drawbacks of more communication and scheduling overhead in this case. Efficiency drops for all methods (excluding random search) when using 64 GPUs in parallel. This indicates that at this point drawbacks of adding more workers to the problem grow, resulting in sub-optimal resource usage. For datasets and models comparable in size to the ones from NATS-Bench, it is therefore recommended to use a maximum of 32 workers for the NAS loop to keep parallel efficiency above $\approx 0.75$, i.e., ensuring better resource exploitation. If practitioners are comfortable with parallel efficiency of only $\approx 0.6$, even up to 64 GPUs can be used, but it should be noted that in this case a large portion of compute resources is spent on communication and scheduling tasks. In response to **RQ2**, the resource allocation investigations on a single GPU show that it is in general advisable to increase the parallelism of the NAS loop to achieve a decreased total runtime. In addition to the wasted resources by idling the GPU when running only a single trial, the overhead created by frequent checkpointing and re-loading of configurations increases the runtime by a large factor. This holds true even when adjusting for GPU utilization by adjustment of the batch size (see. Fig. 3), though the effect becomes notably smaller. The same conclusions can be made when running data-parallel training on multiple GPUs – at least some degree of parallelism of the NAS loop should be ensured. If it is the aim to get a strong anytime performance, it is advisable to increase the resources available to the single training loops. However, the main takeaway message of this study is that also small levels of NAS parallelism (just $2 - 4$ concurrently running trials) deliver a good trade-off between fast overall runtime and strong anytime performance. This takeaways message is consistent both on small and the large resource budgets, on small and large datasets and models and on various types of GPUs.

## 6  Conclusion and Future Work

In this work, the optimized and sub-optimal resource allocation for the three early-stopping based NAS methods ASHA, BOHB, and HB has been explored. The scalability on three image classification datasets with CNN models from NATS-Bench has been obtained, and recommendations on how many resources to use in total, and how to balance them between single trials and overall NAS loop has been provided. Our analysis contributes to the field of AutoML by providing guidance on how to allocate their devices when running a study with NAS algorithms with the ultimate goal of achieving fast runtimes. For future work, it is of interest to develop strategies that perform adaptive re-allocation of hardware resources, in similar fashion as HB does from a temporal perspective.

## 7  Broader Impact Statement

This work presents a study on optimal resource allocations for NAS methods. While our evaluations are initially computationally expensive, the findings in this paper help researchers to run their NAS algorithms more efficiently, resulting in less wasteful use of resources and lower energy consumption.

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

## A  Untruncated Plots Including Uncertainty

For reasons of readability, the plots in the main part of the paper were truncated and omitted uncertainty measurements. In the following, these plots are shown without truncation and including standard deviation as uncertainty measurement. Figure 7 is the untruncated version of Fig. 2, Fig. 8 of Fig. 4, and Fig. 9 of Fig. 4.

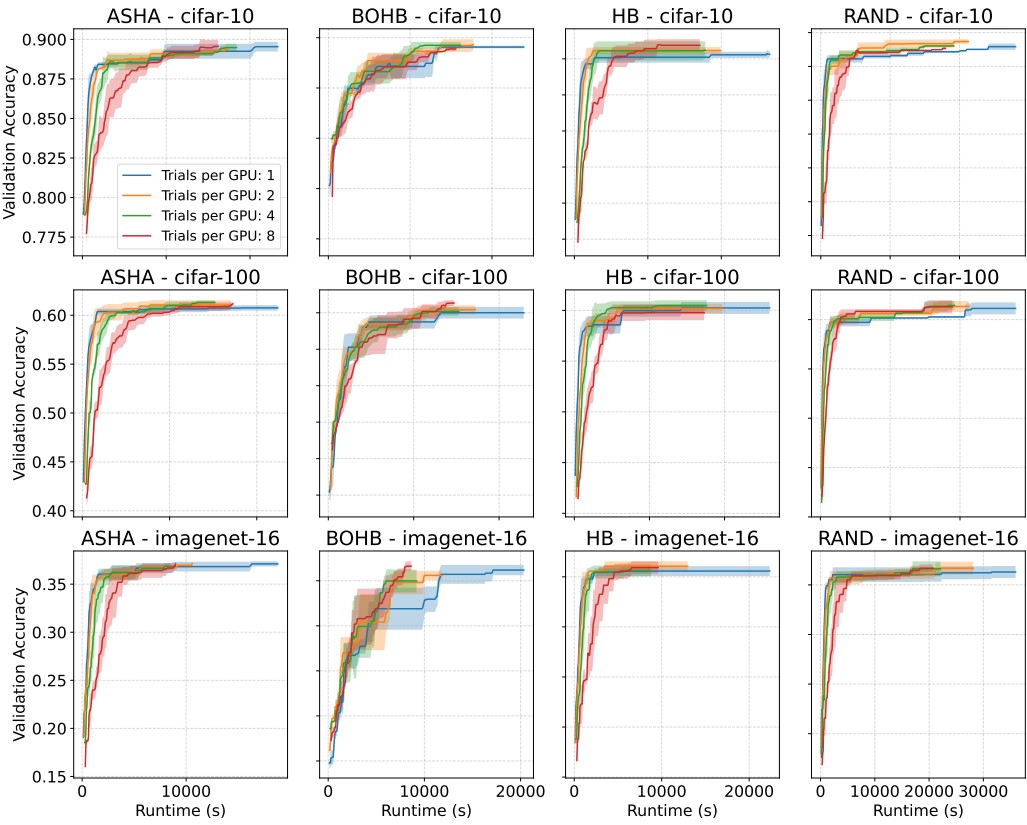

Figure 7: Multiple trials on a single GPU, showing validation accuracy over runtime. Results are averaged over three different seeds.

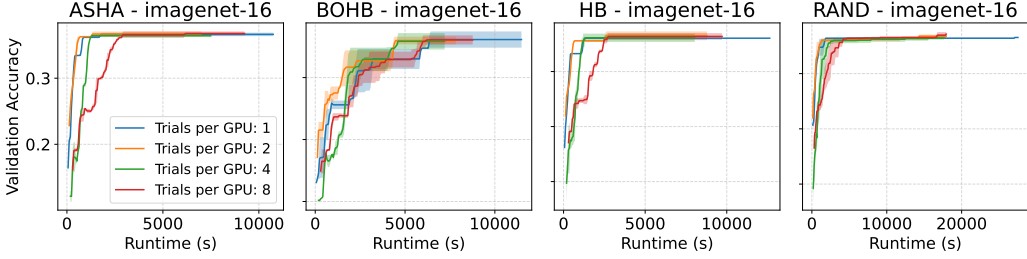

Figure 8: Multiple trials on a single GPU with inverse batch scaling according to the number of trials, showing validation accuracy over runtime. The results are averaged over three different seeds.

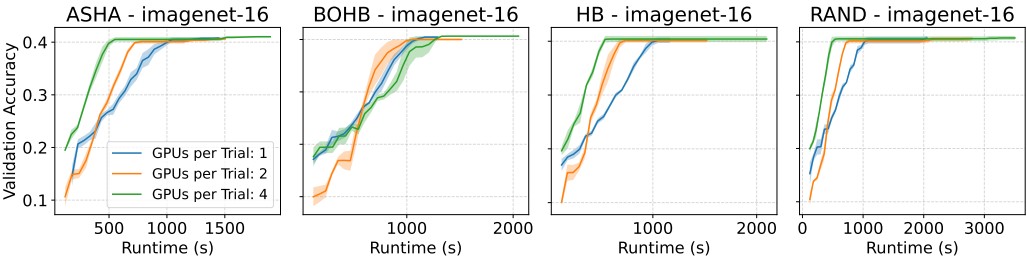

Figure 9: Distributing trials across **multiple** GPUs, showing validation accuracy over runtime. The results are averaged over three different seeds.

## B   Scaling Plots

This section presents the scaling plots from Sec. 4.1 and Fig. 1 in more extensive way, i.e. the plots are split to show the parallel effiency on the small scale (1-8 GPUs) and on the large scale (16-64 GPUs), see Fig. 10.

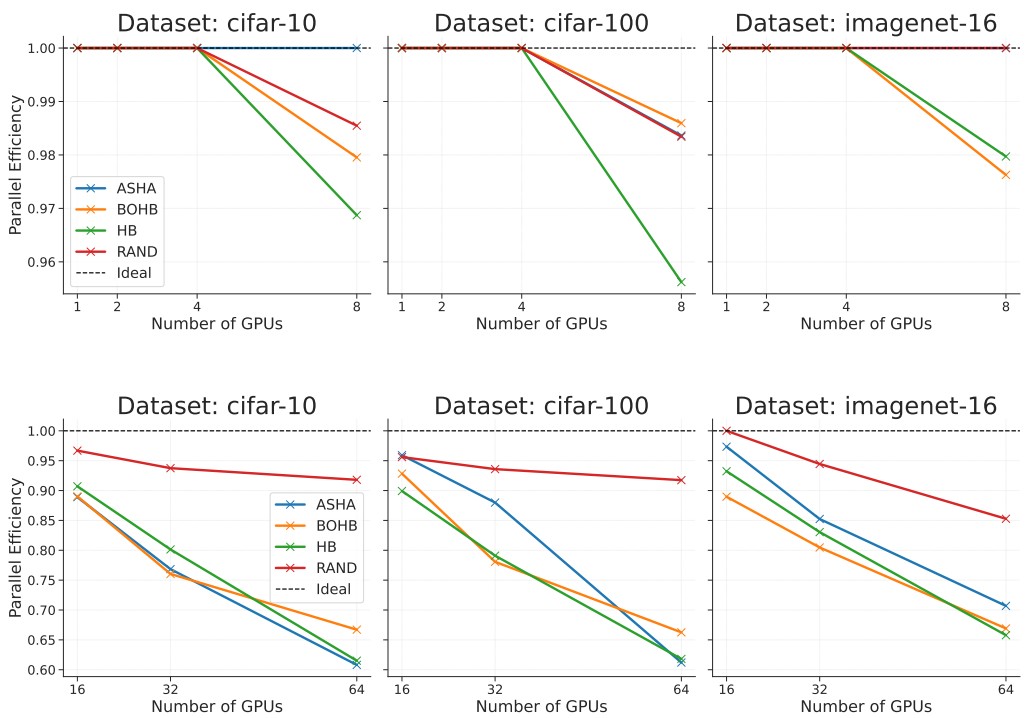

Figure 10: Scalability comparison: Parallel efficiency $E_G$ over the number of GPUs (Top: 1-8, Bottom: 16-64), based on the runtime until 128 samples are evaluated. $E_G = 1$ denotes the ideal case.

