# OpenReview forum: "Optimal Resource Allocation for Early Stopping-based Neural Architecture Search Methods"
_automl.cc/AutoML/2023/Conference — AutoML 2023 MainTrack_

### Official Review · Reviewer_XDAv · 2023-04-12

**Potential Impact On The Field Of Automl Rating:** 2
**Technical Quality And Correctness Rating:** 2
**Clarity Rating:** 3

**Summary Of Contributions:**

The paper studies the impact of using variable numbers of workers and GPUs on the speed of NAS within NATS-Benchmark. More specifically it studies two main research questions: 1) Efficiency of NAS as a function of number of GPUs used, and 2) Impact on the speed and accuracy of NAS when using various numbers of trials per GPU. From the results: 1) Efficiency has a decreasing trend when using more GPUs compared to using a single GPU. 2) NAS is generally faster when using more workers on a GPU, but if there are too many workers, then the NAS time can start increasing. There is no significant impact on accuracy. The authors study the question further by controlling for the amount of simultaneously processed data and find that the main reason hindering efficient resource usage are the switching and reloading processes between trials.

**Actions Required To Increase Overall Recommendation:**

It would be useful to conduct evaluation using more settings (especially various types of GPU but also other benchmarks would be useful) so that the guidance is more likely to generalize to new settings.

**Clarity:**

The paper is generally clear and it is simple to read. There are quite a few typos and simple grammar mistakes, so these should be fixed. Examples include:
* L119 training is is especially
* L151 nestorv momentum for
* L194 allocated to a single trials
* L211 these kinds of decision can


**Overall Review:**

Positives
* The paper studies the problem of resource allocation from multiple perspectives: 1) scaling across multiple GPUs, 2) scaling workers within 1 GPU
* The paper is relatively easy to read
* The topic of the paper may be interesting in case there are any surprising findings that would challenge how resources are allocated
* Multiple common HPO algorithms (and hence of interest to researchers) are evaluated

Negatives:
* Any specific recommendations in the paper are likely to depend on the exact combination of benchmark and GPU used. Because only NATS-Benchmark and one type of GPU was evaluated it is hard to use any of the more specific suggestions such as using 32 GPUs in parallel and not more. To make the specific recommendations more general (and hence useful) more settings/combinations should be considered, but it can give mixed results.
* A large part of the take-aways about efficiency in RQ1 depends on using threshold of 0.75. Various people may prefer different thresholds and the results can be quite different when considering different setups, so claims that we should use up to 32 can be unjustified. The results generally show that efficiency decreases as more GPUs are used.
* While in some cases average across 3 runs is reported, no results show standard deviations. These would be especially useful in Table 1 to see if the points raised in the discussion actually hold or are simply due to noise.
* The high-level findings of the paper are not particularly surprising as they essentially suggest that “it is advisable to increase the parallelism of the NAS loop to achieve a decreased total runtime”.


**Potential Impact On The Field Of Automl:**

The paper can be seen as a case study showing the impact of allocating resources in various ways on NATS-Benchmark when using a specific GPU, with the high-level findings likely to generalize. However, the general findings agree with earlier findings (existing small-scale studies mentioned in section 2.4) so in this sense the paper does not bring particularly surprising findings. As a result I expect only limited impact.

**Reproducibility (Optional):**

I had a quick look at the code and it seems to provide all details that would be needed for reproducing the results - including scripts to run the experiments.


**Review Confidence:**

4: You are confident in your assessment, but not absolutely certain. It is unlikely, but not impossible, that you did not understand some parts of the submission or that you are unfamiliar with some pieces of related work.

**Review Rating:**

5: Borderline Leaning Reject: Technically sound paper where reasons to reject nonetheless outweigh reasons to accept. Please use sparingly.

**Review Summary:**

Based on the paper it is hard to take specific and justified insights about how many workers or GPUs one should use when facing a new setup. The current paper is more of a case-study of the impact of using different numbers of workers on the specific setup considered.


**Technical Quality And Correctness:**

The evaluation considers a rather limited setup of NATS-Benchmark on a specific type of GPU so it is hard to draw specific conclusions that would tell us more about how to allocate resources. In this sense the claims in the paper are not sufficiently sound as the level of empirical evaluation does not enable us to make the more general claims in the paper. More specifically one would expect that the identified numbers in the paper strongly depend on what GPU is used - most GPUs (e.g. 2080 Ti or 1080 Ti) would likely have those thresholds lower than A100). On the other hand the general findings are relatively obvious and may have been identified in earlier papers (although without rigorous studies - based on section 2.4 of this paper). Some evaluations are averaged over 3 seeds, but e.g. Table 1 would benefit from including standard deviations to see if we can be confident about the behaviour we see there.

---

> ### Author Response · Authors · 2023-04-27
> **Reply to Reviewer XDAv**
>
> We thank the reviewer for the assessment of our work and the provided feedback. We address all concerns raised by the reviewer below.
>
> > The paper can be seen as a case study showing the impact of allocating resources in various ways on NATS-Benchmark when using a specific GPU, with the high-level findings likely to generalize. However, the general findings agree with earlier findings (existing small-scale studies mentioned in section 2.4) so in this sense the paper does not bring particularly surprising findings. As a result, I expect only limited impact.
>
> We thank the reviewer for this comment. We are aware that small-scale studies on the scalability and resource allocations of NAS methods have been performed before, however, these are different from our evaluations in the sense that the type of hardware that is used (e.g. just CPUs), the number of workers (GPUs) are different and the models trained are much smaller. These types of evaluations do not transfer to experimental setups currently used by researchers, which is where our study provides guidance. We have added a sentence to Section 3.2 for clarification and hope that the reviewer now agrees with the impact of this work.
>
> > The evaluation considers a rather limited setup of NATS-Benchmark on a specific type of GPU so it is hard to draw specific conclusions that would tell us more about how to allocate resources. In this sense, the claims in the paper are not sufficiently sound as the level of empirical evaluation does not enable us to make the more general claims in the paper. More specifically one would expect that the identified numbers in the paper strongly depend on what GPU is used - most GPUs (e.g. 2080 Ti or 1080 Ti) would likely have those thresholds lower than A100).
>
> We agree with the reviewer that performing experiments on different types of GPUs, in particular less powerful GPUs than A100s would bolster our findings and thank him for this suggestion. We have therefore decided to perform additional experiments on clusters featuring NVIDIA V100 and AMD MI50 GPUs. We chose those types of GPUs as they are more prevalent in current HPC and cluster systems than, e.g., commercial NVIDIA Ti GPUs. The results are presented in the new Subsection 4.5 and Figure 6 of the revised paper. The results of these experiments support our findings that increasing the parallelism of the NAS loop while decreasing the resources of the single trials leads to lower runtimes.
>
> > On the other hand the general findings are relatively obvious and may have been identified in earlier papers (although without rigorous studies - based on section 2.4 of this paper).
>
> We would like to emphasize the main finding of our work that increasing the parallelism of the NAS loop (which at the same time means decreasing the parallelism of the single trials, as the total number of workers in a setting stays the same) reduces the total runtime of the NAS loop.
>
> > Some evaluations are averaged over 3 seeds, but e.g. Table 1 would benefit from including standard deviations to see if we can be confident about the behavior we see there.
>
> We agree that showing standard deviations is important for quantifying the reliability of the results and thank the reviewer for pointing this out. As the lines in the plots are close together (see e.g. Fig. 3), we have decided for omitting them in the plots in the main paper. However, we do now include the untruncated figures with the standard deviations as uncertainty measurement in Appendix A1 of the revised manuscript, whereas Table 1 of the revision now includes standard deviations.
>
> > The paper is generally clear and it is simple to read. There are quite a few typos and simple grammar mistakes, so these should be fixed. Examples include:
> L119 training is is especially
> L151 nestorv momentum for
> L194 allocated to a single trials
> L211 these kinds of decision can
>
> We have fixed these typos in the revised paper.
>
> > Any specific recommendations in the paper are likely to depend on the exact combination of benchmark and GPU used. Because only NATS-Benchmark and one type of GPU was evaluated it is hard to use any of the more specific suggestions such as using 32 GPUs in parallel and not more. To make the specific recommendations more general (and hence useful) more settings/combinations should be considered, but it can give mixed results.
>
> We agree with the reviewer that to make more general recommendations it is important to perform experiments in different settings. We have therefore added two more types of experiments: one using older NVIDIA V100 GPUs and AMD MI 50 GPUs to assess the impact of the type of hardware and one using larger models from a ResNet50 search space. It should be noted that the results of the new evaluations support our previous conclusions (see Subsections 4.4 and 4.5 and Figure 6).

---

> > ### Author Response · Authors · 2023-04-27
> > **Continuation of Reply to Reviewer XDAv**
> >
> > > A large part of the take-aways about efficiency in RQ1 depends on using threshold of 0.75. Various people may prefer different thresholds and the results can be quite different when considering different setups, so claims that we should use up to 32 can be unjustified. The results generally show that efficiency decreases as more GPUs are used.
> >
> > As the reviewer points out correctly, different thresholds might be suitable for different scenarios. We have therefore added a sentence to Section 5 to emphasize that users that are comfortable with a parallel efficiency of 0.6 can use up to 64 GPUs, but more computational resources will be spent on communication and scheduling instead of computation in that case.
> >
> > > While in some cases average across 3 runs is reported, no results show standard deviations. These would be especially useful in Table 1 to see if the points raised in the discussion actually hold or are simply due to noise.
> >
> > The standard deviations are now reported in Table 1. As they are small, they do not influence the final results.
> >
> > > The high-level findings of the paper are not particularly surprising as they essentially suggest that “it is advisable to increase the parallelism of the NAS loop to achieve a decreased total runtime”.
> >
> > See our previous answer to the first question.
> >
> > > It would be useful to conduct evaluation using more settings (especially various types of GPU but also other benchmarks would be useful) so that the guidance is more likely to generalize to new settings.
> >
> > In addition to the new Subsection 4.5 with experiments on various types of GPUs, we have also added new experiments on a different NAS search space; a custom ResNet50 model trained on the complete imagenet-1k dataset(see Subsection 4.4 and Figure 6). The results of these new experiments confirm that our main findings do not depend on the type of hardware used.
> >
> > We hope to have addressed all concerns raised by the reviewer and would be happy to answer any follow-up questions.

---

> > ### Comment · Reviewer_XDAv · 2023-05-08
> > **Response**
> >
> > Thanks for the additional explanations and for conducting the additional experiments using various GPUs. My main worry is that these have not been used for also answering RQ1 where specific guidance about efficiency is given (e.g. use up to 32 GPUs if you want efficiency >0.75). One could expect that the answer to RQ1 depends on what GPU you use, which could lead to challenges with what message the paper gives.
> >
> > The additional clarifications have been helpful, especially highlighting that the overall number of workers remains the same in RQ2. I see some value in studying this as people may not be fully utilising GPUs (due to running one trial on one GPU). But on the other hand if a GPU is well-utilised with a single trial on it (e.g. by using the max batch size that fits there for speed - which can be very common to do), then it may not be useful to put multiple trials there. So a limitation of this study is that the findings likely only apply to problems that do not fully utilise GPU, making the findings problem-dependent. So overall the guidance about speedups can apply only to certain (limited) cases.
> >
> > Overall I increase my rating by one level only (borderline leaning reject), due to the limitations outlined here.

---

### Official Review · Reviewer_oBzf · 2023-04-12

**Potential Impact On The Field Of Automl Rating:** 4
**Technical Quality And Correctness Rating:** 3
**Clarity Rating:** 3

**Summary Of Contributions:**

This paper tackles the problem of parallel compute resource allocation in the context of Neural Architecture Search (NAS) for early stopping-based algorithms. The authors distinguish two assignment choices in the presence of multiple GPUs: 1) assign more resources for speeding up individual configuration training, and 2) assign more resources for NAS procedure for increased parallel configuration sampling. Furthermore, using a unified benchmark, NATS-Bench, they investigate the optimal allocation trade-off on three datasets in a setup of varying numbers of GPUs available (from 1 to 64) of four baselines. Empirical results show that by optimizing the allocation, a speed-up of factor ~2-5 can be achieved.

**Actions Required To Increase Overall Recommendation:**

- Improve the clarity of the paper.
- Improve results presentation.
- Present paper contributions in a more convincing manner.
- Show that optimized allocation is transferable.
- Show large-scale experiments.
- Present takeaway messages more clearly.

**Clarity:**

The motivation is strong and the potential impact on the field is big. However, there are some points that need clarification (see Technical Quality and Correctness Rating). Moreover, the final take-away message is not entirely clear and gives no definitive answer to the posed research questions.

Please consider moving the research questions from the introduction to the experiments section such that a reader does not need to scroll up to orient themself and replace them with clear paper contributions.

Please consider working on the presentation of the plots and explain how they were truncated.

**Overall Review:**

Strengths:
- The paper is well-motivated and bears potential practical significance.
- The evaluation protocol is standardized.
- The appropriate baselines are chosen for empirical evaluation.
- The scaling experiment, measuring parallel efficiency is very interesting.
- The insights about batch size scaling for better GPU utilization are valuable.
- The future work direction, especially adaptive resource allocation, seems very promising.

Weaknesses:
- The checkpointing mechanisms (for a single GPU experiment) is not entirely clear, along with its overhead influencing overall runtime.
- The presentation of some of the figures could be improved to better visualize discussed speed-ups. Maybe displaying initial speed-ups on one plot and the final on another would declutter them.
- The resource allocation scheme has been optimized for a single search space parametrization, it would be recommended to demonstrate that this carries to different experiment setups and can be reused for different problems.
- It would be great to see experiments for large models.
- Clear takeaway messages are missing, to guide researchers in their experimental setup.

**Potential Impact On The Field Of Automl:**

The usage of parallel compute resources for Neural Architecture Search is becoming standard both for research and practical considerations. Hence, the insights into how optimally allocate those resources can be influential for the field.

**Review Confidence:**

3: You are fairly confident in your assessment. It is possible that you did not understand some parts of the submission or that you are unfamiliar with some pieces of related work.

**Review Rating:**

6: Borderline Leaning Accept: Technically sound paper where reasons to accept outweigh reasons to reject. Please use sparingly.

**Review Summary:**

The authors investigate the optimal resource allocation for early stopping-based algorithms. They motivate their research very well and pose interesting research questions. The evaluation protocol has some minor flaws when it comes to clarity and presentation. The empirical results and discussion do not present clear takeaway messages for future research which downplays the potential impact of the paper. As I opt for borderline leaning reject, I encourage to incorporate the feedback to improve the overall paper quality and increase the score.

**Technical Quality And Correctness:**

The empirical study follows a standardized evaluation protocol using open-source libraries for ease of comparison. However, the reviewer would like to clarify:
1. In the related work describing HyperBand (L78-79), "HB waits for all trials to reach a certain threshold in time before applying Successive Halving" which results in idling workers. Isn't it the case that if there are idling workers, another SH bracket will be spawned? E.g. let's say we have 64 workers, and the first bracket would have 27 configurations evaluated for one epoch, then the remaining 37 workers would not wait for the first 27 to finish, but another bracket with 9 configurations for 3 epochs will start, and so on.
2. The early stopping-based approaches revolve around stopping and continuing some of the configurations training, do you always train from scratch or checkpoint and load the model? E.g. in case you evaluated a configuration for 1 epoch already when we promote it for 3 epochs, is it trained for 3 (from scratch) or for 2 epochs?
3. Follow-up to the above, is this continuation cost reflected on the x-axes in paper figures?
4. The experiments with parallel runs on a single GPU. Does that imply constant switching between configurations training as only one configuration can be trained at once? E.g. in case we have scheduled 9 configurations for 3 epochs, do we take one, train for one epoch, checkpoint, train another configuration for one epoch, checkpoint, and so on? It might be interesting to see what is the overhead of checkpointing and loading models in this scenario.
5. It is recommended to include MOBSTER or any multi-fidelity BO method that implements fantasizing the results of pending evaluations.

---

> ### Author Response · Authors · 2023-04-27
> **Reply to Reviewer oBzf**
>
> We thank the reviewer for this assessment of our paper and the feedback. We address all concerns raised by the reviewer below.
>
> > In the related work describing HyperBand (L78-79), "HB waits for all trials to reach a certain threshold in time before applying Successive Halving" which results in idling workers. Isn't it the case that if there are idling workers, another SH bracket will be spawned? E.g. let's say we have 64 workers, and the first bracket would have 27 configurations evaluated for one epoch, then the remaining 37 workers would not wait for the first 27 to finish, but another bracket with 9 configurations for 3 epochs will start, and so on.
>
> The reviewer is correct. If there are enough free workers to start the computation of a new bracket, the HB algorithm would do so. However, HB still needs to wait for all trials of a bracket to reach a certain threshold before it can decide on which trials to stop. This is the main difference to the ASHA algorithm, which does so on a rolling basis. We improved the clarity by modifying the statement that HB leads to idling workers to reflect the point raised by the reviewer (see Section 2.1) in the revised manuscript.
>
> > The early stopping-based approaches revolve around stopping and continuing some of the configurations training, do you always train from scratch or checkpoint and load the model? E.g. in case you evaluated a configuration for 1 epoch already when we promote it for 3 epochs, is it trained for 3 (from scratch) or for 2 epochs?
>
> There is a checkpoint that saves the current state of each training (including model weights) every 10 epochs, whereasdecisions on which trainings to continue and which to stop are also made in the same 10 epoch intervals. Therefore, when a configuration is promoted to the next stage, it would be reloaded from the checkpoint and, hence, is not trained from scratch.
>
> > Follow-up to the above, is this continuation cost reflected on the x-axes in paper figures?
>
> The runtime plotted in the figures is the whole runtime of the NAS optimization process, so it includes the overhead from the checkpointing and loading of these checkpoints (note that as mentioned above, there is no re-training from scratch).
>
> > The experiments with parallel runs on a single GPU. Does that imply constant switching between configurations training as only one configuration can be trained at once? E.g. in case we have scheduled 9 configurations for 3 epochs, do we take one, train for one epoch, checkpoint, train another configuration for one epoch, checkpoint, and so on? It might be interesting to see what is the overhead of checkpointing and loading models in this scenario.
>
> It is correct that on a low level, the GPU can just compute a single problem at a time. Therefore, when running parallel trials on a single GPU, “context switching” is used. This means that the GPU constantly switches between different threads, but at a much higher frequency (e.g., after the computation of a single batch of a model) than the employed checkpointing intervals (e.g., every 10 epochs). Unfortunately, information from the hardware providers (such as NVIDIA) on how this is performed under the hood is sparse. But, in our experiments, we measure the whole runtime of the NAS loop (including any overhead created by this context switching) and our conclusions suggest that running more trials in parallel on a single GPU leads to a faster runtime. We therefore assume these overheads to be small. This information, where context swichtig is used in the case of sharing a single GPU is included to Section 3.1 of the revised manuscript.
>
> > It is recommended to include MOBSTER or any multi-fidelity BO method that implements fantasizing the results of pending evaluations.
>
> We thank the reviewer for this suggestion. However, the Ray Tune framework for the evaluations to avoid implementation-specific disturbance factors used in our work and the MOBSTER framework are unfortunately not compatible out of the box.  Hence, we sincerely apologize for not being able to include this suggestion due to the limited time for this rebuttal.
>
> > The motivation is strong and the potential impact on the field is big. However, there are some points that need clarification (see Technical Quality and Correctness Rating). Moreover, the final take-away message is not entirely clear and gives no definitive answer to the posed research questions.
>
> We thank the reviewer for this positive feedback, and have reformulated Section 5 to directly answer the posed research questions and also clearly formulate the main takeaway message.
>
> > Please consider moving the research questions from the introduction to the experiments section such that a reader does not need to scroll up to orient themself and replace them with clear paper contributions.
>
> As suggested we have added our main contribution to the introduction of the revised manuscript.

---

> > ### Author Response · Authors · 2023-04-27
> > **Continuation of Reply to Reviewer oBzf**
> >
> > > Please consider working on the presentation of the plots and explain how they were truncated.
> >
> > The plots were truncated in the sense that the x-axis was shortened towards the end (as the blue lines run for a long time) and the y-axis was truncated on the bottom. For clarity, we have added the untruncated Figures to Appendix A1 and mentioned the truncation more clearly in the revised manuscript (see Section 4.2).
> >
> > > The checkpointing mechanisms (for a single GPU experiment) is not entirely clear, along with its overhead influencing overall runtime.
> >
> > In our experiments, we measure the whole runtime of the NAS optimization process, therefore, the runtime measurement of the trials on a single GPU includes any overhead. It is advisable from our conclusions to increase the parallelism of the NAS loop.
> >
> > > The presentation of some of the figures could be improved to better visualize discussed speed-ups. Maybe displaying initial speed-ups on one plot and the final on another would declutter them.
> >
> > We thank the reviewer for suggesting to declutter the speed-up plots to help with readability. As requested, we have split the Figures into initial and final speed-ups and included them in the Appendix B of the revised manuscript, as space in the main paper is sparse.
> >
> > > The resource allocation scheme has been optimized for a single search space parametrization, it would be recommended to demonstrate that this carries to different experiment setups and can be reused for different problems.
> >
> > We agree with the reviewer that demonstrating the resource allocations schemes in more experimental setups would be helpful to ensure the generalizability of the findings, and therefore, added an evaluation on a ResNet search space with the (full size) imagenet-1k dataset (see Section 4.4 and Figure 6). Additionally, we include evaluation on more HPC systems, equipped with NVIDIA V100 and AMD MI50 GPUs (see Section 4.5).
> >
> > > It would be great to see experiments for large models.
> >
> > As mentioned in our previous answer, we have added a study on a ResNet search space. The models sampled from this search space are roughly two orders of magnitude larger than the models from NATS-Bench (e.g., 30M vs 0.3M parameters, respectively).
> >
> > > Clear takeaway messages are missing, to guide researchers in their experimental setup.
> >
> > We have modified Section 5 to answer the research questions directly and state the main takeaway message (to increase the parallelism of the NAS loop at the cost of parallelism of the single trials) more clearly.
> >
> > > Improve the clarity of the paper.
> > Improve results presentation.
> > Present paper contributions in a more convincing manner.
> > Show that optimized allocation is transferable.
> > Show large-scale experiments.
> > Present takeaway messages more clearly.
> >
> > We believe to have addressed all the points raised in our previous answers and are happy to answer any follow-up questions.

---

> > > ### Comment · Reviewer_oBzf · 2023-05-07
> > > **Increased score**
> > >
> > > Thank you for the clarifications and for the additional content!

---

### Official Review · Reviewer_Kwnc · 2023-04-13

**Potential Impact On The Field Of Automl Rating:** 3
**Technical Quality And Correctness Rating:** 4
**Clarity Rating:** 4

**Summary Of Contributions:**

The authors experimentally investigate the impact of parallelism of the NAS loop on the runtime and quality of the outcome using three common early-stopping based methods: Hyperband, Asynchronous successive halving (ASHA), and Bayesian optimization hyperband (BOHB). In particular, two research questions were investigated: how many resources should be allocated to a single NAS task, and how many resources should be allocated to a single NAS trial (evaluating a single architecture configuration). The authors conclude that using up to 32 GPUs per NAS task is advisable for general early-stopping based NAS methods, and that having a smaller number of trials per GPU (e.g., 1-4 trials per GPU) gives stronger anytime performance in terms of validation accuracy than setting 8 trials per GPU.



**Actions Required To Increase Overall Recommendation:**

It would be nice if the authors could address the three points mentioned in "clarity".


**Clarity:**

How exactly does the global gradient "degrade" (line 129) with a smaller number of devices? Would appreciate it if the authors could clarify that more.

The fact that E_G in Equation (3) is bounded above by 1 implies that T_G is bounded below by 1/G T_1. Is this due to overhead created by communication and scheduling? Though it may be common sense for domain experts, it would still be nice if this could be explicitly explained.

Possible typo: "substantial" rather than "substational" on line 109.


**Overall Review:**

The experiments are well-designed and thoroughly conducted and the conclusions drawn from the experiments are reasonable. The insights in this paper could provide useful guidance for ML practitioners to allocate resources for NAS.

The paper is also well-written. The motivation, related work, and experiments are clearly explained.

**Potential Impact On The Field Of Automl:**

This paper provides guidance for ML practitioners in allocating GPU resources for early-stopping based NAS approaches. Others are likely to benefit from this paper's insights and cite this paper for its contributions.


**Reproducibility (Optional):**




**Review Confidence:**

3: You are fairly confident in your assessment. It is possible that you did not understand some parts of the submission or that you are unfamiliar with some pieces of related work.

**Review Rating:**

8: Accept: Technically sound paper with major impact and strong evaluation, with perhaps some minor flaws.

**Review Summary:**

The methodology seems sound, the explanations are clear, and the insights are beneficial to a wide range of practical situations. Therefore I recommend accept.


**Technical Quality And Correctness:**

The experiments conducted and conclusions drawn seem to be sound and of high quality.

---

> ### Author Response · Authors · 2023-04-27
> **Reply to Reviewer Kwnc**
>
> We thank the reviewer for the assessment of our work and the provided feedback. We address all concerns raised by the reviewer below.
>
> > How exactly does the global gradient "degrade" (line 129) with a smaller number of devices? Would appreciate it if the authors could clarify that more.
>
> We agree with the reviewer that the sentence induces confusion and thank her/him for pointing this out. It is only possible to use the learning rate scaling technique for mitigation of the degradation of generalization performance if the number of devices is small. If the number of devices is large, even using this technique cannot help avoid the degradation of generalization performance. We have reformulated this sentence in the revised manuscript.
>
> > The fact that E_G in Equation (3) is bounded above by 1 implies that T_G is bounded below by 1/G T_1. Is this due to overhead created by communication and scheduling? Though it may be common sense for domain experts, it would still be nice if this could be explicitly explained.
>
> Indeed, T_G is bounded below by 1/G T_1. Since T_1 is the runtime of the problem on a single worker and T_G is the runtime of the problem on G workers, this lower bound represents the perfect scaling case. In some cases,  this value can go below the threshold (which would mean an E_G value of larger than 1), which is referred to as superlinear speed-up. We have added a sentence on this in the revised paper.
>
> > Possible typo: "substantial" rather than "substational" on line 109.
>
> We have fixed this typo in the revised paper.
>
> > It would be nice if the authors could address the three points mentioned in "clarity".
>
> We hope to have addressed all concerns raised by the reviewer and would be happy to address any follow-up questions by the reviewer.

---

### Official Review · Reviewer_UhWx · 2023-04-17

**Potential Impact On The Field Of Automl Rating:** 3
**Technical Quality And Correctness Rating:** 4
**Clarity Rating:** 3

**Summary Of Contributions:**

This paper performs an empirical study that compares early stopping-based NAS methods: HB, ASHA, and BOHB. The main question that it seeks to answer is how to best allocate resources, both for the NAS experiment overall (Q1) and for a single trial (Q2).

For Q1, the paper looks at parallel efficiency vs. number of GPUs allocated, showing that with up to 32 GPUS, the parallel efficiency is above 0.75, but for 64 GPUs, the efficiency drops to 0.6-0.65. For Q2, the paper suggest increasing parallelism in order to get a reduced runtime.

**Actions Required To Increase Overall Recommendation:**

* I think if this paper performed a more systematic evaluation and provided clearer results and takeaways (esp for RQ2), then it would have increased impact.
* (Copied from above) One question I expected to get an answer to is the following: suppose I had a total compute budget along with a instantaneous compute budget, how should I trade off adaptivity vs wall time?

**Edit** I've increased the score after reading responses. Thank you to the authors for the prompt response.

**Clarity:**

The authors should include error bars on all plots. As it currently stands, the plots don't convey much information.

**Overall Review:**

Strengths:
* The main strength of this paper is an empirical comparison of the methods, which can help NAS practitioners. The answer to RQ1 was clear and conclusive and perhaps provides an insight that was not previously known

Weaknesses:
* For RQ2, is the main takeaway to increase parallelism as much as possible? This could be made clearer throughout. Although this is not too surprising, it is still nice for practitioners to know.
* The plots should have error bars.
* One question I expected to get an answer to is the following: suppose I had a total compute budget along with a instantaneous compute budget, how should I trade off adaptivity vs wall time?

**Potential Impact On The Field Of Automl:**

This paper provides some nice advice to practitioners on what settings to use when applying HB, ASHA, and BOHB.

**Review Confidence:**

4: You are confident in your assessment, but not absolutely certain. It is unlikely, but not impossible, that you did not understand some parts of the submission or that you are unfamiliar with some pieces of related work.

**Review Rating:**

6: Borderline Leaning Accept: Technically sound paper where reasons to accept outweigh reasons to reject. Please use sparingly.

**Review Summary:**

Although the paper gives a nice empirical comparison of three early stopping-based methods, I think the impact at this point is still relatively small.

**Technical Quality And Correctness:**

I did not see issues with correctness except that the plots should show error bars.

---

> ### Author Response · Authors · 2023-04-27
> **Reply to Reviewer UhWx**
>
> We thank the reviewer for the assessment of our work and the provided feedback. We address all concerns raised by the reviewer below.
>
> > For RQ2, is the main takeaway to increase parallelism as much as possible? This could be made clearer throughout. Although this is not too surprising, it is still nice for practitioners to know.
>
> To clarify: the main takeaway is to increase the parallelism of the NAS loop as much as possible at the cost of the parallelism of the single trials. We agree that this was not made clear enough in the paper and thank the reviewer for pointing this out. To make this more clear, we have modified Section 5 to answer the research questions from the introduction and present the main takeaway. Additionally, we highlight this finding also already in the introduction (Section 1) after formulating the research questions.
>
> > The plots should have error bars.
>
> We agree that showing error bars is important for quantifying the reliability of the results and thank the reviewer for pointing this out. As the lines in the Figures are close together (see e.g. Fig. 2), we have decided for omitting them in the plots in the main paper. However, we do now include the untruncated plots with error bars in  Appendix A1.
>
> > One question I expected to get an answer to is the following: suppose I had a total compute budget along with a instantaneous compute budget, how should I trade off adaptivity vs wall time?
>
> We would like to ask the reviewer to elaborate on this question, especially on what is meant with an “instantaneous compute budget” and with “adaptivity”. We will try to answer the question and include the answer in the paper once we have received this.
>
> > I think if this paper performed a more systematic evaluation and provided clearer results and takeaways (esp for RQ2), then it would have increased impact.
>
> We have extended our evaluation in a more systematic fashion to include larger models (Subsection 4.4) as well as different types of GPUs (Subsection 4.5, Figure 6). In addition to our modification of Section 5 to highlight our main takeaway, we hope to have addressed all concerns raised by the reviewer and would be happy to address any follow-up questions by the reviewer.

---

> > ### Comment · Reviewer_UhWx · 2023-05-05
> > **Response**
> >
> > > We would like to ask the reviewer to elaborate on this question, especially on what is meant with an “instantaneous compute budget” and with “adaptivity”. We will try to answer the question and include the answer in the paper once we have received this.
> >
> > By "adaptivity", I mean the parallelism of NAS (lower parallelism = higher adaptivity). By "instantaneous compute budget", I mean the maximum parallelism at any given time. By total compute budget, I mean total resources used overall across time (think total AWS cost, for instance).

---

> > > ### Author Response · Authors · 2023-05-07
> > > **Response to the Reviewers Question**
> > >
> > > We thank the reviewer for the clarification. From our understanding, the question is aimed at how to balance the total amount of computational resources (e.g. AWS cost) with the amount of resources used at the same time (e.g. number of GPUs allocated concurrently).
> > >
> > > The answer to this question is a combination of RQ1 ("How many resources should be allocated in total for a NAS?") and RQ2 ("How many resources should be allocated for each NAS trial?") from our paper. Our findings in regard to RQ1 suggest that using a total number of 32 workers in parallel is favorable for maintaining a high computational efficiency (>0.75). Using more than 32 workers leads to lower computational efficiency and therefore to the waste of resources. Our findings in regard to RQ2 suggest that as much parallelism of the NAS loop as possible is favorable for reducing the runtime. The answer to the question is therefore: It would be best to allocate the *total available compute budget* in such a fashion that an *instantaneous compute budget* of 32 workers (i.e., GPUs) can be used in parallel and at the same time select the *adaptivity* in such a fashion that the parallelism of the NAS loop is high (as many trials as possible on those 32 workers in parallel).
> > >
> > > Modifications to the paper are unfortunately no longer possible, but we believe to have addressed this already in our revised paper in the conclusion in lines 311 to 318 (new text marked in red).

---

### Review · Reproducibility_Reviewer_LZmS · 2023-04-18

**Completeness Of Code And Dataset Supplement Rating:** 4
**Usability And Ease Of Reproducibility Rating:** 4
**Actions Required To Increase The Reproducibility And Overall Recommendation:** N/A

**Completeness Of Code And Dataset Supplement:**

The code and dataset supplements are complete and sufficient.

**Overall Reproducibility Review:**

The experiments and code are reproducible. The README is clear and easy to follow, which makes it easier for others to reproduce the experiments and build upon the code in future research.

**Review Confidence:**

4: You are confident in your assessment, but not absolutely certain. It is unlikely, but not impossible, that you did not understand some parts of the submission or that you are unfamiliar with some pieces of the code or data.

**Review Rating:**

8: Accept, all aspects of this are reproducible with minor effort.

**Review Summary:**

As a reproducibility reviewer, I recommend this paper for acceptance.

**Summary Of Necessary Code And Dataset Supplement:**

This paper investigates the optimal allocation of a fixed amount of parallel workers for conducting Neural Architecture Search (NAS) using early stopping-based NAS methods. The authors test HyperBand, Asynchronous Successive Halving, and Bayesian Optimization HyperBand in tuning models in parallel from one GPU to up to 64 GPUs. The experiments are performed on CIFAR10, CIFAR100, and ImageNet16 on NATS-Bench. The authors conclude that selecting the appropriate number of parallel evaluations can accelerate the NAS process by factors of ≈ 2 − 5 while preserving the test set accuracy compared to non-optimal resource allocations.

**Usability And Ease Of Reproducibility:**

README is well-organized and easy to follow. I can reproduce the results using the provided code.

---

### Author Response · Authors · 2023-04-27
**General rebuttal reply**

We would like to thank all reviewers for their valuable feedback and suggestions to improve the paper. We have significantly extended our experimental evaluations, now including runs on two more types of GPUs (NVIDIA V100 and AMD MI50). Moreover, another NAS search space, where sampled models are trained on the complete imagenet dataset (see Subsections 4.4 and 4.5 and Figure 6), is included to the revised manuscript. These evaluations are inline with the previous conclusions and main takeaway, which has been that an increase of the NAS parallelism is favorable, even at the cost of decreasing the parallelism of the single trials. Due to space constraints, the Figures with lines including standard deviations as uncertainty measurements are now included in Appendix A1. As it took us some time and a large amount of computing resources to run the new experiments, four days were needed for updating the paper. All major additions to the paper are highlighted in red.

---

> ### Author Response · Authors · 2023-05-01
> **End of Discussion Period Approaching**
>
> Dear reviewers,
>
> as the end of the author-reviewer discussion period is approaching, we are reaching out to resolve any additional questions you may have regarding our responses to your comments and the revised paper. We want to ensure that all your concerns are thoroughly addressed before the deadline.
>
> We thank you for your time and consideration.

---

> > ### Author Response · Authors · 2023-05-05
> > **End of Second Discussion Period**
> >
> > Dear Area Chair,
> > dear Reviewers,
> >
> > we would kindly remind you that the second phase of the discussion period ends this Monday. We are in your service if any additional clarification to our responses is needed.
> >
> > We thank you for your time and consideration.